# Assessment and Imaging of Intracellular Magnesium in SaOS-2 Osteosarcoma Cells and Its Role in Proliferation

**DOI:** 10.3390/nu13041376

**Published:** 2021-04-20

**Authors:** Concettina Cappadone, Emil Malucelli, Maddalena Zini, Giovanna Farruggia, Giovanna Picone, Alessandra Gianoncelli, Andrea Notargiacomo, Michela Fratini, Carla Pignatti, Stefano Iotti, Claudio Stefanelli

**Affiliations:** 1Department of Pharmacy and Biotechnology, University of Bologna, 33, 40127 Bologna, Italy; emil.malucelli@unibo.it (E.M.); giovanna.farruggia@unibo.it (G.F.); giovanna.picone2@unibo.it (G.P.); stefano.iotti@unibo.it (S.I.); 2Department of Biomedical and Neuromotor Sciences, Alma Mater Studiorum, University of Bologna, 33, 40126 Bologna, Italy; maddalena.zini@unibo.it (M.Z.); carla.pignatti@unibo.it (C.P.); 3INBB—Biostructures and Biosystems National Institute, 00136 Rome, Italy; 4Elettra—Sincrotrone Trieste, 34149 Trieste, Italy; alessandra.gianoncelli@elettra.eu; 5Institute for Photonics and Nanotechnologies, Consiglio Nazionale delle Ricerche, 00156 Rome, Italy; andrea.notargiacomo@ifn.cnr.it; 6Institute of Nanotechnology-CNR c/o Physics Department at ‘Sapienza’ University, 00185 Rome, Italy; michela.fratini@gmail.com; 7IRCCS Fondazione Santa Lucia, 00179 Rome, Italy; 8Department for Life Quality Studies, Alma Mater Studiorum, University of Bologna, 47921 Rimini, Italy; claudio.stefanelli@unibo.it

**Keywords:** magnesium, osteosarcoma, cell cycle, mTOR

## Abstract

Magnesium is an essential nutrient involved in many important processes in living organisms, including protein synthesis, cellular energy production and storage, cell growth and nucleic acid synthesis. In this study, we analysed the effect of magnesium deficiency on the proliferation of SaOS-2 osteosarcoma cells. When quiescent magnesium-starved cells were induced to proliferate by serum addition, the magnesium content was 2–3 times lower in cells maintained in a medium without magnesium compared with cells growing in the presence of the ion. Magnesium depletion inhibited cell cycle progression and caused the inhibition of cell proliferation, which was associated with mTOR hypophosphorylation at Serine 2448. In order to map the intracellular magnesium distribution, an analytical approach using synchrotron-based X-ray techniques was applied. When cell growth was stimulated, magnesium was mainly localized near the plasma membrane in cells maintained in a medium without magnesium. In non-proliferating cells growing in the presence of the ion, high concentration areas inside the cell were observed. These results support the role of magnesium in the control of cell proliferation, suggesting that mTOR may represent an important target for the antiproliferative effect of magnesium. Selective control of magnesium availability could be a useful strategy for inhibiting osteosarcoma cell growth.

## 1. Introduction

Magnesium is an essential nutrient with a wide range of metabolic, structural and regulatory functions [1]. Despite its presence in several types of food, the absorption and elimination of magnesium might be easily hindered by several factors. Even after its absorption, many substances may increase the excretion of magnesium in the kidneys and cause reduced plasma levels, such as excessive alcohol intake, diuretics, coffee, salt, sugar and excess fat [2].

It is estimated that from 2.5% to 15% of the world’s population experiences some form of hypomagnesemia. Magnesium deficiency is frequently observed in industrialized countries. The US National Health and Nutrition Examination Survey (NHANES) stated that approximately one-half of all American adults have an inadequate intake of magnesium [3].

Magnesium is the fourth most abundant mineral and the second most abundant intracellular divalent cation in the body. Approximately 50% of magnesium can be found in bone, and approximately 50% is inside body tissue cells and organs, while less than 1% is found in the blood. A great deal of evidence shows that magnesium acts primarily as a signalling element in cell metabolism, and the concept that Mg^2+^ is simply an electrolyte is obsolete [4,5]. In fact, magnesium plays a crucial role in many cellular processes, such as energy metabolism [6,7], protein and DNA synthesis [8,9] and several studies show that magnesium content directly correlates to proliferation in normal and transformed cells [10,11]. 

Intracellular magnesium homeostasis is primarily maintained by the ubiquitously expressed ion channel transient receptor potential melastatin (TRPM)7, which is a member of the transient receptor potential (TRP) family, possessing both ion channel and kinase activities [12]. TRPM7 and its homolog TRPM6 are strictly associated with intracellular signalling. Upon mitogen stimuli, cells are able to increase their intracellular magnesium content, most likely activating its influx. In contrast, magnesium deprivation inhibits DNA and protein synthesis and promotes cell growth arrest. A wide range of literature provides evidence about the essential role of magnesium in the transduction of proliferative signals. This could be explained by considering that protein kinases are strictly dependent on the complex of magnesium and ATP (MgATP^2−^), which is the biologically active form in all living organisms [5]. As regards the role of magnesium in the control of cell proliferation, Rubin [4] postulated that the binding of growth factors to their membrane receptors causes a perturbation of the plasma membrane and consequently a release of the magnesium bound to membrane phospholipids. This release leads to a significant rise in the cytosolic free magnesium concentration, allowing Mg^2+^ to displace other cations from the ATP complexes, increasing the active form of MgATP^2−^.

Despite the evident link between magnesium and cell proliferation, the role of magnesium in cancer cells is scarcely documented and often contradictory [13,14,15]; in particular, the role of magnesium in primary bone tumours has not yet been examined. It is worth noting that in physiological conditions, magnesium plays an important role in bone metabolism [16,17,18] and can influence osteoblast and osteoclast differentiation, affecting bone growth and remodelling [19,20]. This work aims to study the role of magnesium in the growth of cancer SaOS2 osteosarcoma cells, monitoring the cellular concentration and compartmentalization of the cation by means of an advanced cellular imaging technique, as well as the effect of magnesium on specific signal transduction pathways.

## 2. Materials and Methods

### 2.1. Reagents

All reagents were Ultrapure grade and, unless otherwise specified, were from Merck-Millipore. 

Dulbecco’s Phosphate-Buffer Saline (DPBS) without Ca^2+^ and Mg^2+^ (8 g L^−^^1^ NaCl, 0.2 g L^−^^1^ KCl, 0.2 g L^−^^1^ Na_2_HPO_4_, 0.2 g L^−^^1^ KH_2_PO_4_, pH 7.2) was prepared in doubly distilled water. The fluorescent probe DCHQ5 was synthesized as previously reported [21] and was dissolved in dimethyl sulfoxide (DMSO) to a final concentration of 1 mg mL^−^^1^. Aliquots were kept in the dark at 4 °C.

Foetal Bovine Serum (FBS, Euroclone, Milan, Italy) was dialyzed by using the Spectra/Por 4 Molecular Porous Dialysis Membrane (Spectrum, Austin, TX, USA) against Puck Buffer (NaCl 136.9 mM, KCl 5.4 mM, NaHCO_3_ 4.2 mM and d-glucose monohydrate 4.2 M) plus EDTA for 2 days, and only Puck Buffer for the last 3 days. Calcium content in dialyzed FBS (dFBS) was restored by adding CaCl_2_ at a final concentration of 1.8 mM, and dFBS was sterilized by filtering through a 0.45 µm pore size membrane filter. 

### 2.2. Cell Culture 

The human osteosarcoma cell line SaOS-2 (American Type Culture Collection, Manassas, VA, USA) was cultured at 37 °C and 5% CO_2_ in MEM medium (Invitrogen, Carlsbad, CA, USA), supplemented with 2 mM l-Glutamine, 10% FBS, 1000 units mL^−1^ penicillin and 1 mg mL^−1^ streptomycin. The cells were seeded at 10^4^ cell/cm^2^ in complete MEM and after 24 h the medium was substituted by the custom-made medium MEM w/o Mg^2+^ (Invitrogen, CA, USA), and where needed, MgCl_2_ was added at 1 mM concentrations. The medium was supplemented with 2 mM l-Glutamine, 10% FBS, 1000 units mL^−1^ penicillin and 1 mg mL^−1^ streptomycin.

To synchronize cells in G0/G1 phase and reduce intracellular magnesium content, cells were cultured in the medium containing 0.5% dFBS in the absence of magnesium for 24 h. Then, in order to stimulate cell proliferation, the cells were grown in a medium containing 5% dFBS in the presence or absence of 1 mM MgCl_2_. To determine the rate of cell proliferation, viable cells were counted after 24 h and 48 h by using a Bürker hemocytometer in the presence of erythrosine 0.1% in PBS.

### 2.3. Flow Cytometric Assays 

Flow cytometric assays were performed on an Epics Elite flow cytometer (Beckman Coulter, Brea, CA, USA) equipped with an Argon Ion laser tuned at 488 nm. 

Cell cycle. To perform the cell cycle analysis, cells were fixed by 70% ice-cold ethanol and left at −20 °C overnight. After centrifugation pellets were resuspended in an appropriate volume (0.5–2 mL) of staining solution (DPBS, 5 µg mL^−^^1^ Propidium Iodide (PI) and 10 µg mL^−^^1^ DNAse free RNase A). Samples were incubated in the dark for 30 min at 37 °C and analysed by acquiring the PI red fluorescence on a linear scale at 600 nm. Data analysis is performed using the software program “ModFit” (Verity, Carrollton, TX, USA).

p27^Kip1^ induction. Detached cells were washed 2 times from the growth medium in DPBS by centrifuging at 240 *g* for 10 min. The samples were then fixed with 3% paraformaldehyde at room temperature for 15 min. To remove any residual formaldehyde, samples were washed 2 times in PBS-glycine 0.1 M, followed by two washes in DPBS-BSA 1% performed to block nonspecific sites. A solution 1:9 of DPBS-ethanol (70%) was added in order to permeabilize the cell membranes and samples are maintained at −20 °C for 3 min. Samples were then washed 3 times in DPBS-BSA 1% by centrifuging at 240 *g* for 5 min. and marked with a rabbit primary antibody anti-p27^Kip1^ under stirring at 4 °C overnight. The samples were then washed in DPBS, and marked with a secondary antibody FITC conjugated, diluted 1:1000 at room temperature for 1 h. Finally, to verify the p27^Kip1^ expression levels in the function of the cell cycle, the samples were counterstained for DNA content by PI 5 µg mL^−1^ and analysed by flow cytometry. FITC green fluorescence is collected at 525 nm on a logarithmic scale and PI red fluorescence at 600 nm on a linear scale.

### 2.4. Lactate Dehydrogenase Assay

To verify the effect of magnesium deprivation on cell viability, released lactate dehydrogenase (LDH) activity was assayed in the culture medium. Briefly, 2 mL of medium were centrifuged at 4000 *g* for 10 min. The supernatant was preserved and constitutes the sample. Sequentially, 1.325 mL of phosphate buffer 0, 1 M at pH 7, 50 μL of sodium pyruvate 23 mM and 50 μL of NADH 14 mM dissolved in TRIS 0.1 M at pH 7 were added to the cuvette. The reaction starts after the addition of 100 μL of the sample. The absorbance was then measured at 340 nm and at intervals of 1 min against a blank prepared with 100 μL of fresh medium.

### 2.5. Western Blotting

The level of protein expression at the indicated time points was evaluated by Western blotting, as previously described [22]. Protein samples were run in 6% (for mTOR) or 15% (for LC3 and p27^kip1^) SDS-polyacrylamide gels. Polyclonal primary antibodies of rabbit anti-mTOR, anti-phospho-mTOR, anti-p27^kip1^, anti-LCRA and anti-LCRB (Cell Signaling Beverly, MA, USA) were diluted 1:1000. The secondary antibody anti-rabbit IgG was diluted 1:2500 in PBS with 3% of skimmed milk powder. The intensity of the bands was evaluated with the densitometric software GelPro Analyzer 3.0 (Media Cybernetic, Rockville, MD, USA). In graphs, band intensity was normalized to the loading control β-actin. 

### 2.6. Magnesium Determination by DCHQ5 Spectrofluorimetric Assay

Detached cells were washed 3 times in DPBS at 240 *g* for 10 min, counted and resuspended at 10^6^ cell mL^−1^, and stored at −20 °C until the spectrofluorimetric analysis. Total intracellular magnesium was assessed on sonicated cell samples by using the fluorescent chemosensor DCHQ5, according to Sargenti et al. [21]. Magnesium concentration was normalized for the amount of cells mL^−1^ used during analysis, and reported as nmoles/10^6^ cells. To obtain the mM concentration of magnesium, the detected nmoles are divided by the cell volume (µL) calculated according to Malucelli et al. [23].

### 2.7. Synchrotron Based X-ray Microscopy

To perform X-ray microscopy [24], the cells were plated on a 200 nm-thick silicon nitride membrane window (Silson UK), grown for 24 h with 5% dFBS in the presence or absence of magnesium, and then dehydrated and fixed by chemical fixing: after two washes in ammonium acetate 100 mM, they were immersed in methanol/acetone 1:1 for 2 min at −20 °C and then air-dried.

The scanning transmission X-ray microscopy (STXM) and the X-ray fluorescence microscopy (XRFM) measurements were carried out at the beamline TWINMIC at Elettra Synchrotron (Trieste, Italy) [25]. The dehydrated cells were carefully examined with an optical microscope and selected following the criteria of integrity, dimensions, and distance from other cells.

A Fresnel zone plate focused the incoming beam (1475 eV), monochromatized by a plane-grating monochromator, to a circular spot of about 600 nm in diameter. The sample was transversally scanned in the zone plate focus pixel per pixel and in steps of 500 nm. At each step, the fluorescence radiation intensity was measured by eight silicon drift detectors (active area 30 mm^2^) concentrically mounted at a 20° grazing angle with respect to the specimen plane, at a detector-to-specimen distance of 28 mm [26]. Simultaneously, the transmitted intensity T was measured by a fast-readout electron-multiplying low-noise charge-coupled device (CCD) detector through an X-ray–visible light converting system [27]. Zone plate, sample, and detectors were in a vacuum, thus avoiding any absorption and scattering by air.

Five STXM images were acquired on whole cells with a step size of 500 nm. In sequence, XRFM and simultaneously, STXM were carried out with a range of 6–8 s dwell time per pixel, depending on the cell size. The total acquisition time was in the range of 6–10 h (field of view of at least 20 μm × 20 μm, and spatial resolution 500 nm). The measurement of I0 was made by acquiring 25 points and repeating the measure five times. Atomic Force Microscopy (AFM) measurements were performed on selected cells before and after XRFM and STXM measurements.

Thereby, maps of magnesium molar concentration can be calculated using the algorithm developed by Malucelli et al. [24]. 

### 2.8. Statistical Analysis 

The experiments were repeated at least three times, and the values were reported as mean ± standard deviation. One-way ANOVA analysis was also performed and values of *p* < 0.05 were taken to be statistically significant.

## 3. Results

### 3.1. Effects of Magnesium Deprivation on Intracellular Magnesium Content

In order to study the effects of magnesium deficiency in human osteosarcoma SaOS-2 cells, the cells were firstly synchronized in G0/G1 phase with a reduced intracellular magnesium content by culturing them for 24 h in a medium without magnesium, containing 0.5% dFBS. Then, the medium was replaced with a medium containing 5% dFBS in the presence or absence of 1 mM MgCl_2_, and grown for 24 and 48 h.

To evaluate the intracellular total magnesium, the fluorescent chemosensor DCHQ5 was used. It is a diaza-crown-hydroxyquinoline that allows the assessment of intracellular total magnesium in a much lower number of cells than compared to other techniques and to other commercial dyes [21].

Following the Rubin model which postulates a release of membrane-bound magnesium and a consequent increase in the MgATP required by the protein kinases after mitogenic stimulation [10], the total intracellular amount of the cation was measured 24 h after the addition of serum in the culture medium. Cells grown for 24 h in the presence or absence of magnesium were lysed by sonication and analysed by spectrofluorometry. Figure 1 shows that in cells grown in a medium without magnesium, the total intracellular content of the ion was about 50% with respect to cells cultured in the presence of magnesium, ranging from 31 to 16.7 nmol/10^6^ cells, respectively. Thus, considering the SaOS-2 volume [23], the intracellular magnesium content was 13.7 mM in cells grown with magnesium and 8.4 mM in cells grown without magnesium.

The dynamic of intracellular magnesium involves changes in its total amount and in cellular compartmentalization. Therefore, to understand the role of magnesium in signal transduction pathways linked to cell proliferation, it is important to evaluate not only its intracellular content but also its spatial distribution. To address this goal, we used an analytical approach of cellular imaging that combines techniques that are not widely utilized in biological studies [24], i.e., atomic force microscopy (AFM), scanning transmission X-ray microscopy (STXM) and X-ray fluorescence microscopy (XRFM). AFM allowed us to measure the thickness of the analysed cells. STXM records the light not absorbed (and then transmitted) by the X-ray irradiated sample, providing information about the local density and producing bidimensional images of the sample. XRFM generates X-ray fluorescence spectra of the cells, whose elaboration allowed us to draw the elemental map of the cell, which is complementary to the transmission map. Figure 2A shows the images of a single SaOS-2 cell obtained by these techniques.

By combining STXM e XRFM images of a single cell, the elemental distributions expressed as weight fractions could be obtained. Malucelli et al. [24] proposed an algorithm which allowed us to combine the weight fraction map, obtained by STXM e XRFM analysis, with AFM data, providing a new elemental distribution map that merged local elemental composition and morphological information (volume). With this approach, it was possible to obtain a gross estimate of the molar concentration map in different zones inside a single cell [24,28]. Hence, the SaOS-2 cells synchronized in G0/G1 phase with reduced intracellular magnesium were stimulated to proliferate by adding 5% dFBS in the presence or absence of 1 mM MgCl_2_. The cells were grown for 24 h on a silica frame and then fixed and analysed by AFM and X-ray microscopy. Figure 2B shows the magnesium maps evaluated as weight fraction and molar concentration of the cells grown in the presence or absence of magnesium. The analysis showed that magnesium concentration was estimated to be 87 mM in cells grown in the presence of MgCl_2_ and decreased to 26 mM in magnesium-deprived cells, confirming the differences reported in Figure 1. In the reported maps, the magnesium concentration can be visually appreciated by a colour scale which goes from a very low signal in blue to a very high signal in red. In cells stimulated to proliferate in the presence of magnesium, the weight fraction map indicates a quite homogeneous distribution of magnesium within the cells, whereas the concentration map shows some “islets” with very high concentration. Furthermore, the central part of the cells shows a low concentration of magnesium, because this is the thickest part of the cells [24]. Interestingly, the cells maintained without magnesium show a somewhat more homogeneous intracellular distribution of the ion without high-concentration zones, and the red and yellow pixels (higher concentration) mainly remain confined in the area corresponding to the plasma membrane.

### 3.2. Effects of Magnesium Deficiency on Cell Growth and Cell Death

The effects of magnesium deficiency on the proliferation of human SaOS-2 osteosarcoma cells were investigated. The cells were firstly synchronized in G0/G1 phase with a reduced intracellular magnesium content, and afterward were grown for 24 and 48 h in a medium containing 5% dFBS in the presence or absence of 1 mM MgCl_2_ as described. At the indicated time points, the cells were counted.

Figure 3A shows that cell proliferation was significantly decreased in cells grown in the absence of MgCl_2_. In detail, the numbers of cells grown in the absence of MgCl_2_ were 62% at 24 h and 53% at 48 h of those grown in the presence of 1 mM MgCl_2_. Considering the SaOS-2 doubling time (36–38 h), the variations in proliferation are better highlighted at 48 h. However, following Rubin’s hypothesis [10,29] we focused on early events and thus we also took into account the cell viability at 24 h.

In order to evaluate the effect of magnesium deficiency on cellular viability, the released LDH activity was measured in a culture medium. Figure 3B shows that the deprivation of MgCl_2_ did not induce any significant increase in LDH release at 24 and 48 h.

Taken together, these results indicate that magnesium deficiency reduced SaOS-2 cell proliferation without affecting cellular viability.

### 3.3. Effects of Magnesium Deficiency on Cell Cycle Progression

It is well known that different cell types have a different dependence on extracellular magnesium availability for their proliferation [10,11]. Hence, we examined the effect of magnesium availability on SaOS-2 cell cycle progression. As established in our experimental protocol, cells synchronized in G0/G1 phase with a low intracellular magnesium content were stimulated to grow in a medium containing 5% dFBS in the presence and absence of 1 mM MgCl_2_ for 24 h and 48 h.

In cells cultured in the absence of MgCl_2_, the percentage of cells in G0/G1 phase was markedly increased with respect to cells grown in the presence of the ion (Figure 4A), being 78% versus 53% at 24 h, and 84% versus 55% at 48 h (Figure 4B). On the other hand, the percentage of cells in S-phase was significantly lower in magnesium-deprived cells, while the cell distribution in the G2/M phase was not substantially influenced by magnesium deficiency. These results indicate that magnesium deficiency suppresses SaOS-2 cell cycle progression from G1 to S-phase, according to previous studies in kidney cells [30].

One feature of the role of magnesium in the control of cell growth pertains to the modulation of cell-cycle inhibitory proteins such as p27^Kip1^ and p21^Cip1/WAF1^ [31]. We investigated the effect of magnesium deficiency only on the expression of p27^Kip1^ since SaOS-2 cells are p53-null and p21^Cip1/WAF1^ protein is prevalently induced by p53 activation [32].

A significant increase in the amount of p27^Kip1^ protein was observed in magnesium-deprived cells stimulated to proliferate by dFBS addition (Figure 4C), suggesting that magnesium is involved in the regulation of p27^Kip1^ in osteosarcoma SaOS-2 cells. Figure 4D shows the expression of p27^Kip1^ in the function of the cell cycle phase. It is noteworthy that an increase in p27^Kip1^ protein was associated with cells not resident in S-phase, as shown by the absence of green fluorescence corresponding to this phase. Overall, the percentage of p27^Kip1^ positive cells ranges from 26% in Mg absence to 62% in Mg presence.

### 3.4. Effects of Magnesium Deficiency on mTOR Signaling

The activation of mTOR kinase represents a fundamental step in the initiation of protein synthesis and in cell growth. We assessed the expression of mTOR protein and mTOR phosphorylation at serine 2448 (S2448), since phosphorylated mTOR binds Raptor and becomes an active kinase [33]. 

Western blot analysis of the protein extracts from cells grown for 24h in the presence or absence of MgCl_2_ revealed that magnesium deficiency did not alter mTOR protein level, but significantly reduced its phosphorylation at S2448 (Figure 5A).

When nutrients are limited, mTOR in the mTORC1 complex is dephosphorylated and dissociates from the ULK complex, initiating the autophagy process [34], leading to the cleavage of LC3 protein which is considered a reliable marker of autophagy in mammalian cells [35]. However, even in the absence of magnesium, mTOR was found to be mainly dephosphorylated at S2448. Magnesium deficiency apparently did not induce LC3 cleavage and, consequently, did not cause a significant increase in the level of LC3-II protein, the cleaved and lipidated form of LC3 protein (Figure 5B). Treatment with 10 mM chloroquine as a positive control of LC3-II accumulation [36] caused a large increase in LC3-II level as expected in both magnesium-deprived cells and control cells (Figure 5C). 

## 4. Discussion

Magnesium is a cofactor involved in more than 300 metabolic reactions in the body, including protein synthesis, cellular energy production and storage, cell growth and reproduction, and deoxyribonucleic acid and ribonucleic acid synthesis. Magnesium helps to maintain normal nerve and muscle function, cardiac excitability, vasomotor tone, blood pressure, immune system, bone integrity, and blood glucose levels. It also promotes intestinal calcium absorption. Based on its multiple functions within the human body, magnesium has been reported to play an important role in the prevention and treatment of many diseases [37], including cancer [38].

Within the cells, magnesium is present at a very high concentration, usually between 5 mM and 30 mM, and only a very small fraction, about 1 mM or less, is unbound. Some authors have suggested that the ionic form moves among cellular sub-compartments [39]. Nevertheless, magnesium intracellular compartmentalization has not yet been thoroughly elucidated, mainly because of the inadequacy of available techniques to map the intracellular distribution of this cation. 

Knowledge of the intracellular concentration and distribution of the chemical elements in cells may reveal their function in a variety of cellular processes. The biological function of a chemical element in cells not only requires the determination of its intracellular quantity but also of the spatial distribution of its concentration [23,40]. In order to address this problem, we applied the multimodal fusion approach developed by Malucelli et al. [24] which combines synchrotron radiation microscopy techniques with off-line atomic force microscopy, and offers the possibility of achieving a detailed map of the intracellular concentration of magnesium [24,28]. This method requires the implementation of images obtained by AFM, XRFM e STXM and the utilization of a specifically elaborated algorithm, allowing an estimate of the molar concentration map of intracellular magnesium. 

In this way, we observed that magnesium is mainly confined at the plasma membrane in quiescent cells. When cells are stimulated to grow with a medium containing 5% serum and normal magnesium concentration, proliferation starts and magnesium moves toward the inner areas of the cell, consistently with Rubin’s model [10]. In contrast, when SaOS-2 cells were stimulated to grow in the absence of extracellular magnesium, the ion mainly remained confined in the area corresponding to the plasma membrane and the cells were unable to proliferate, as shown by the reduced number of cells and by the accumulation in G0/G1 phase of the cell cycle. 

Many tumour cells proved to be resistant to magnesium deficiency [31,32], even if the reduction of magnesium to a very low level can modify cell cycle distribution in some tumour cell lines [18,31]. SaOS-2 cells appear to be sensitive to magnesium depletion, and the block of the cell cycle is associated with a significant increase in p27^Kip1^ that in these p53-null cells represents the main inhibitor of cyclin-dependent kinases. A similar upregulation of p27^Kip1^ in cancer cell lines grown in magnesium-deficient medium has been reported [41,42].

Magnesium deficiency was also shown to decrease mTOR phosphorylation at serine 2448. Phosphorylation of serine 2448 is associated with mTORC1 complex activation [33,43], whereas hypophosphorylation causes mTORC1 inhibition and can lead to the activation of autophagy [34]. However, we cannot detect any significant change in the amount of LC3-II, the cleaved and lipidated form of LC3 protein, whose increase represents a hallmark of autophagy [35], suggesting that autophagy is not a major mechanism associated with the reduced proliferation of magnesium-deficient cells.

Magnesium has an essential role in the transduction of proliferative signals. Rubin’s theory about the role of magnesium in the control of proliferation [10,29] postulates that the release of membrane-bound magnesium leads to an increase in cytosolic free Mg^2+^ with a consequent increase in MgATP, required by protein kinases. Among the kinases involved in proliferative pathways, mTOR has an unusually high K_m_ for MgATP, about 1 mM. Thus, the MgATP complex is a limiting factor for the activation of mTOR kinase, initiation of protein synthesis and, consequently, the progression of the cell cycle from the G1 phase [10,29]. This consideration, together with our finding that magnesium deprivation causes mTOR hypophosphorylation at S2448, suggests that the antiproliferative effect of magnesium deficiency in osteosarcoma cells is mediated by mTOR. Further work is required to explore the effect of magnesium deficiency on other kinases of the signalling cascades involving mTOR. 

Several epidemiological studies have provided evidence that a correlation exists between dietary magnesium and various types of cancer. In addition, impaired magnesium homeostasis is reported in cancer patients, and frequently complicates therapy with some anti-cancer drugs [38].

High levels of magnesium in drinking water protect against oesophageal and liver cancer, and it is inversely correlated with death from breast, prostate, and ovarian cancers, whereas no correlation existed for other tumours [14]. Dietary magnesium intake has been reported to have a statistically significant nonlinear inverse association with the risk of colorectal cancer. The greatest reduction for magnesium intake was a result of 200–270 mg/day [44]. Another study suggested that increasing the intake of magnesium-containing foods may help reduce the incidence and mortality of primary liver cancer [15].

Interestingly, a recent study found that cancer survivors used dietary supplements at a higher frequency and dose than individuals without cancer, but had an overall lower intake of nutrients from foods [45]. Our results indicate that the control of magnesium availability could be a useful strategy for inhibiting osteosarcoma cell growth, and support the hypothesis that mTOR may represent a target for the antiproliferative effect of magnesium deficiency. 

Magnesium is also important for bone health. Interestingly, SaOS-2 cells display osteoblastic features similar to primary human osteoblastic cells and are often used as a model of osteogenic differentiation [20,46]. A recent review showed how optimal magnesium and vitamin D balance may improve bone metabolism and health outcomes [18]. Optimal magnesium levels contribute to the maintenance of skeletal health [47,48], and our results also suggest a mechanism that may be involved in the effects of magnesium deficiency in normal bone cells. This aspect is worthy of attention since there is a profound lack of awareness of the insufficient intake of magnesium in the population worldwide, and the decrease in magnesium content in processed foods and in newer varieties of grains, fruits, and vegetables poses a further challenge for adequate magnesium consumption. 

## 5. Conclusions

Magnesium is an essential nutrient, but the links between magnesium, cell growth, and carcinogenesis still remain unclear and complex, with conflicting results being reported from many experimental, epidemiological and clinical studies.

It has been proposed that transformation causes a selective loss of the growth regulatory role of Mg^2+^.

In view of the evidence that transformed cells have a diminished capacity to regulate their free Mg^2+^, the effects of Mg^2+^ deprivation on their behaviour were examined. In this study, we examined the effect of magnesium deficiency on the proliferation of SaOS-2 osteosarcoma cells. Magnesium depletion limited the ability of cells to progress in the cell cycle and caused the inhibition of cell proliferation, which was associated with mTOR hypophosphorylation at Serine 2448. In order to map the intracellular concentration and compartmentalization of the cation, an advanced cellular imaging technique using synchrotron-based X-ray techniques was applied. When cell growth was stimulated, magnesium was mainly localized near the plasma membrane in cells maintained in a medium without magnesium, whereas in non-proliferating cells growing in the presence of the ion, high concentration areas inside the cell were observed. 

These results are compatible with Rubin’s theory about the role of magnesium in the control of cell proliferation [29], and indicate that selective control of magnesium availability in osteosarcoma cells could be a useful strategy for inhibiting tumour growth.

## Figures and Tables

**Figure 1 nutrients-13-01376-f001:**
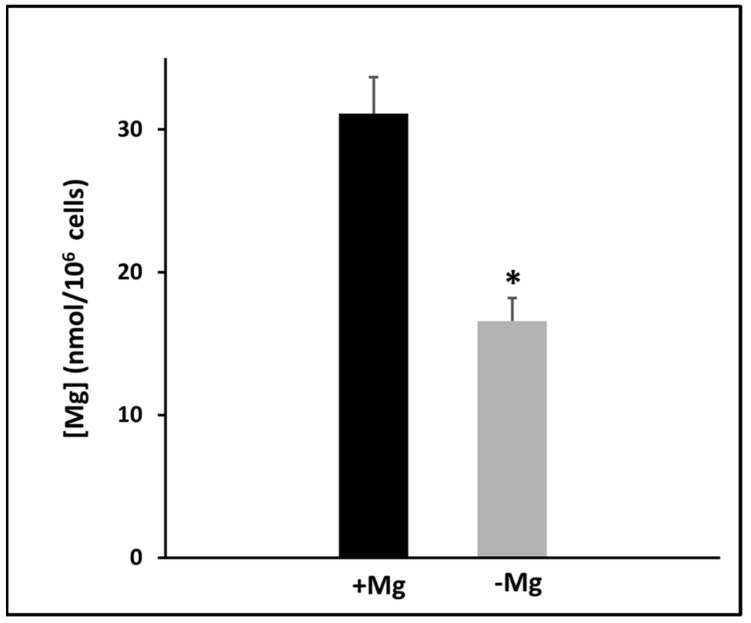
Intracellular content of total magnesium in SaOS-2 cells grown 24 h in the presence (+Mg) or absence (−Mg) of magnesium. Starved cells maintained in magnesium-free medium were stimulated to proliferate by adding 5% dFBS in the presence (+Mg) or absence (−Mg) of 1 mM MgCl_2_. After 24 h, magnesium content was measured by a fluorescent probe. The data are reported as a mean ± SD of three independent experiments. * *p* < 0.01 vs. +Mg.

**Figure 2 nutrients-13-01376-f002:**
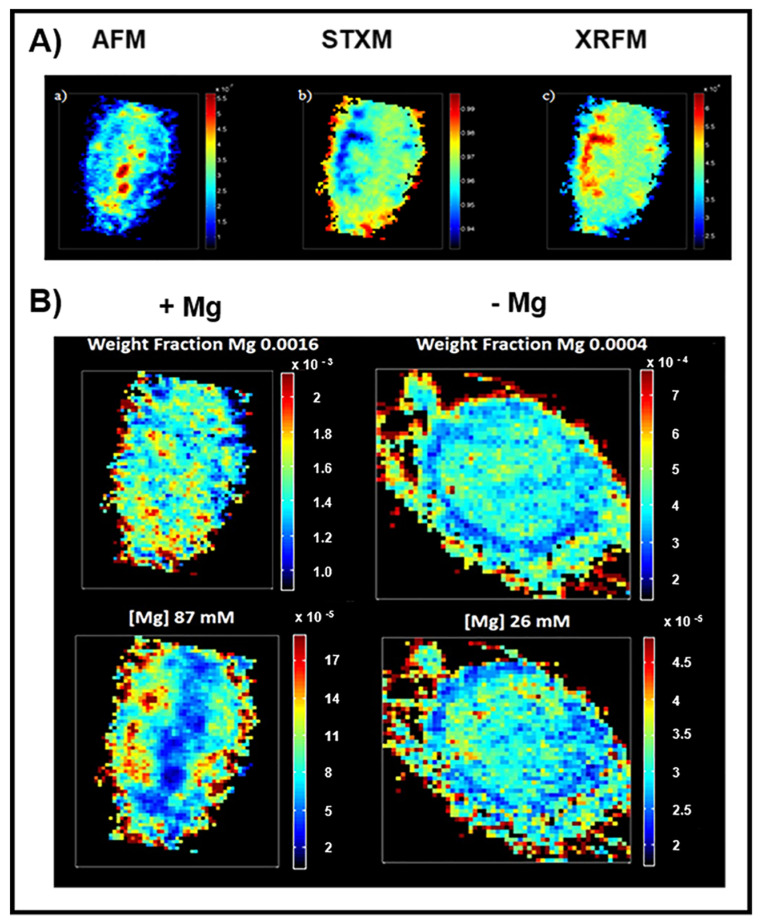
(**A**) Images of a single growing SaOS-2 cell obtained by three different microscopy techniques. (**a**) AFM analysis: the blue (red) colour indicates areas of lesser (greater) thickness; (**b**) STXM analysis: the blue (red) colour indicates areas in which the transmitted radiation is minimal (maximal); (**c**) XRFM analysis: the image is complementary to the STXM image because the incident radiation is more absorbed in the minimum transmission areas causing fluorescence. The blue (red) colour indicates areas of lesser (greater) fluorescence. (**B**) Distribution maps of the magnesium content in SaOS-2 cells cultured 24 h with 5% dFBS in the presence (left) or absence (right) of 1 mM MgCl_2_. At the top, the weight fraction maps are reported. Below, the molar concentration maps are reported.

**Figure 3 nutrients-13-01376-f003:**
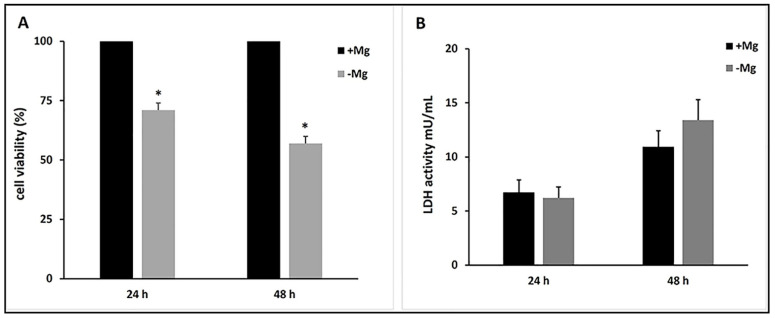
Effect of magnesium deficiency on SaOS-2 proliferation and viability. (**A**) Starved cells were stimulated to proliferate by adding 5% dFBS in presence (+Mg) or absence (−Mg) of 1 mM MgCl_2_. After 24 h and 48 h the cell number was measured. The number of cells grown in the medium including MgCl_2_ was arbitrarily taken as 100%. Data are means ± SD of three independent experiments. * *p* < 0.05 vs. +Mg. (**B**) LDH activity was measured in the culture medium of SaOS-2 cells grown 24 h and 48 h grown in the absence (−Mg) or presence (+Mg) of 1 mM MgCl_2_. The data are reported as a mean ± SD of three determinations.

**Figure 4 nutrients-13-01376-f004:**
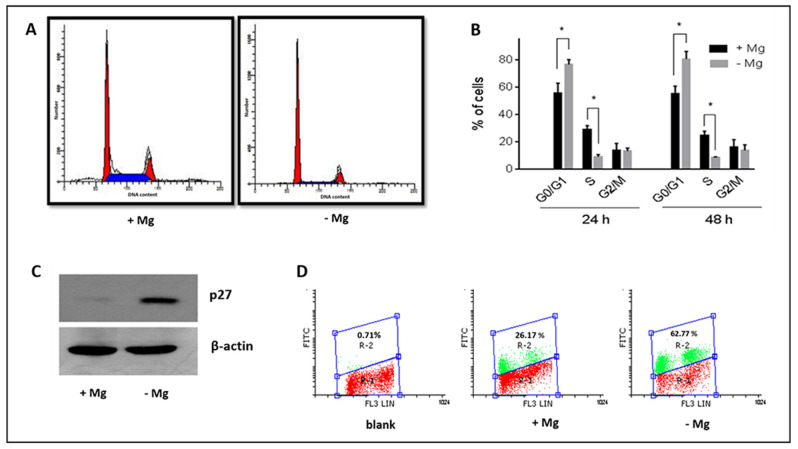
Effect of magnesium on the cell cycle progression of SaOS-2 cells. Starved cells were stimulated to proliferate by addition of 5% dFBS in the presence (+Mg) or absence (−Mg) of 1 mM MgCl_2_ and analysed after the indicated times: (**A**) Typical cell cycle distribution after 24 h from serum addition, determined by flow cytometry. (**B**) percentage of cells in cell cycle phases after 24 h and 48 h; data are means ± SD obtained in three determinations; * *p* < 0.05. (**C**) Western blot analysis of p27^Kip1^ protein in cells grown 24 h in the absence (left) and presence (right) of magnesium. The blot is representative of three experiments. (**D**) Expression of p27^Kip1^ protein at 24 h in the function of cell cycle distribution determined by bi-parametric analysis: PI fluorescence (cell cycle) is shown on the X axis, while FITC fluorescence (p27^Kip1^ protein) is reported on the Y axis.

**Figure 5 nutrients-13-01376-f005:**
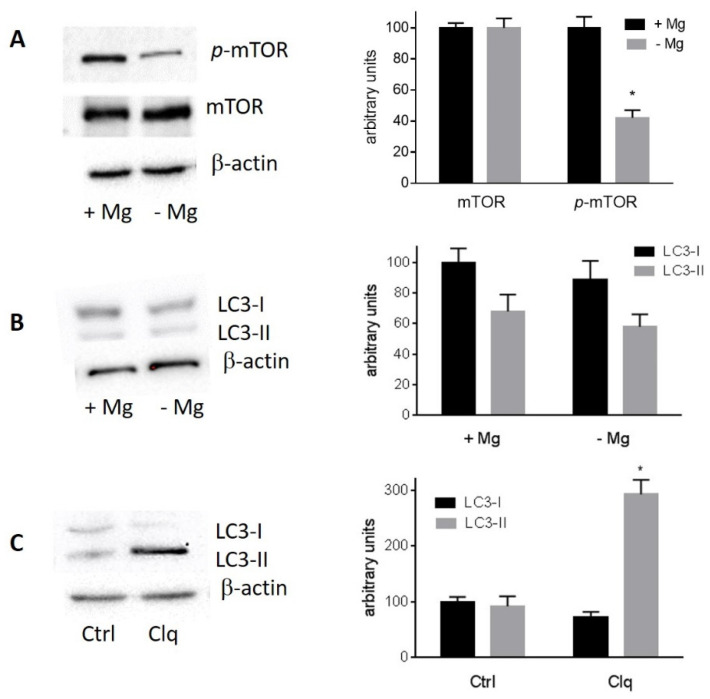
Effect of magnesium deficiency on mTOR level and phosphorylation and LC3 cleavage. Starved SaOS-2 cells were stimulated to proliferate by the addition of 5% dFBS in the presence (+Mg) or absence (−Mg) of 1 mM MgCl_2_. After 24 h, cells were collected for protein analysis by Western blotting: (**A**) Left, Western blot analysis of total mTOR and phosphorylated mTOR (S2448). Right, densitometric analysis; the levels of mTOR and phosphorylated mTOR in the presence of magnesium are arbitrarily taken as 100; data are means ± SD of three determinations, * *p* < 0.05 vs. +Mg. (**B**) Left, Western blot analysis of LC3-I and LC3-II expression in cells grown in the presence or absence of 1 mM MgCl_2_. Right, densitometric analysis; the level of LC3-I in the presence of magnesium is arbitrarily taken as 100; data are means ± SD of three determinations. (**C**) Left, the effect of 10 mM chloroquine (Clq) for 24 h in cells deprived of magnesium is shown as a positive control of LC3-II accumulation. Right, densitometric analysis; the level of LC3-I in control cells is arbitrarily taken as 100; data are means ± SD of three determinations, * *p* < 0.05 vs. control. Similar results were obtained in cells grown in the presence of 1 mM MgCl_2_.

## Data Availability

Not applicable.

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
