# Peer review of "Assessment and Imaging of Intracellular Magnesium in SaOS-2 Osteosarcoma Cells and Its Role in Proliferation"

_nutrients, 2021, doi:10.3390/nu13041376_

Round 1
Reviewer 1 Report
It was my pleasure to revise the manuscript entitled: “Magnesium and osteosarcoma cell proliferation”, presented by Stefanelli C and collaborators. The authors exploited the effects of magnesium deprivation in SaOS2 cell proliferation.
Although I find this work potentially interesting, I have the following concerns:
Major
Since it has been demonstrated that human derived osteosarcoma cell lines display differentiation capacity and growth, invasion, and migration (Mohseny, A., Machado, I., Cai, Y. et al.Functional characterization of osteosarcoma cell lines provides representative models to study the human disease. Lab Invest 91, 1195–1205 (2011), meaning that a single cell model is not enough to explain the pathology, I believe authors should perform the experiments at least using another cell line.
Figure 3A: authors stated that “that cell proliferation was significantly decreased in cells grown in the absence of MgCl2…. the numbers of cells grown in absence of MgCl2 were 62% at 24h”. Since SaOS2 doubling time is reported to be about 43-48 hours, these number could be justified if a percentage of cells died. Which means that magnesium is a fundamental element to restart the cellular metabolism machinery after starvation and that serum alone is not bursting enough. I think this aspect should be deeper investigated and discussed. Also, authors reported there was no autophagy activation, therefore I suggest investigating if other pathways of cell death were activated. I would recommend reporting all the histograms to the 24 hours +Mg (e.g. only set this to 100 and relate all the others to this %), so that it would be clearer whether SaOS2 doubled after 48 hours and it would exacerbate the effect of Mg deprivation. Whether authors referred to viable cells (as they reported in math&meth), they should change the graph name and the text according.
Figure 3B: authors stated that “Figure 3B depicts that the deprivation of MgCl2 did not induce any significant increase of LDH release at 24 h and 48 h”. However, the increase in LDH activity at 48 hours in Mg deprived cells (grey histograms), seems to show the opposite.
For the reasons explained above, I disagree with the conclusion authors stated in the last paragraph sentence: “Taken together, these results indicate that Magnesium deficiency reduced SaOS-2 287 cell proliferation without affecting cellular viability”.
Minors
As author discussed, Magnesium is a key co-factor for protein kinases ATP-dependent. Author only investigated the effect of Mg deprivation on mTOR kinase, but many protein kinases are involved in the regulation of cell proliferation. One of the ways to simultaneously screening the activity of several protein kinases, is to perform a protein kinase antibody array, which consist of a nitrocellulose membrane on which several protein kinase antibodies (against the protein and its phosphorylated forms) are spotted. This will also elucidate how the downstream mTOR signaling targets are regulated.
Author Response
Response to Reviewer 1 Comments
Major
POINT 1: Since it has been demonstrated that human derived osteosarcoma cell lines display differentiation capacity and growth, invasion, and migration (Mohseny, A., Machado, I., Cai, Y. et al.Functional characterization of osteosarcoma cell lines provides representative models to study the human disease. Lab Invest 91, 1195–1205 (2011), meaning that a single cell model is not enough to explain the pathology, I believe authors should perform the experiments at least using another cell line.
RESPONSE 1: Thank you for your suggestion: we changed the title to specify that we chose SaOS2 cells because they represent a widely employed model of human osteosarcoma. We agree that performing experiments on other osteosarcoma cell lines would give more specific information, but this is outside the goal of our study.
POINT 2: Figure 3A: authors stated that “that cell proliferation was significantly decreased in cells grown in the absence of MgCl2…. the numbers of cells grown in absence of MgCl2 were 62% at 24h”. Since SaOS2 doubling time is reported to be about 43-48 hours, these number could be justified if a percentage of cells died. Which means that magnesium is a fundamental element to restart the cellular metabolism machinery after starvation and that serum alone is not bursting enough. I think this aspect should be deeper investigated and discussed.
RESPONSE 2: In our experimental conditions, we repeatedly observed that SaOS2 cells have a doubling time of about 36 hours, data recently confirmed also in 3D cell culture (Picone, G., Cappadone, C., Pasini, A., et al. Analysis of intracellular magnesium and mineral depositions during osteogenic commitment of 3D cultured SaOS2 cells. Int. J. Mol. Sci, 2020, 21(7), 368). We agree that variations in proliferation are better highlighted at 48 hours. However, taking into account the Rubin’s hypothesis on cell proliferation, we also focused on earlier events at 24 hours. We modified the text to clarify this issue (lines 305-308).
POINT 3: Also, authors reported there was no autophagy activation, therefore I suggest investigating if other pathways of cell death were activated. I would recommend reporting all the histograms to the 24 hours +Mg (e.g. only set this to 100 and relate all the others to this %), so that it would be clearer whether SaOS2 doubled after 48 hours and it would exacerbate the effect of Mg deprivation. Whether authors referred to viable cells (as they reported in math&meth), they should change the graph name and the text according.
RESPONSE 3: We agree with the referee. The presentation of results is not clear since we actually performed a viability assay. Accordingly, the title of figure 3A has been changed.
POINT 4: Figure 3B: authors stated that “Figure 3B depicts that the deprivation of MgCl2 did not induce any significant increase of LDH release at 24 h and 48 h”. However, the increase in LDH activity at 48 hours in Mg deprived cells (grey histograms), seems to show the opposite.
RESPONSE 4: Thanks for the careful review. We fully agree: figure 3B is awkwardly wrong. The standard deviation bars were incorrectly reported. We corrected the graph and reported the statistical analysis showing that differences were not significant.
POINT 5: For the reasons explained above, I disagree with the conclusion authors stated in the last paragraph sentence: “Taken together, these results indicate that Magnesium deficiency reduced SaOS-2 287 cell proliferation without affecting cellular viability”.
RESPONSE 5: We are confident that the clarification of the previous points supports our conclusion.
Minors
POINT 6: As author discussed, Magnesium is a key co-factor for protein kinases ATP-dependent. Author only investigated the effect of Mg deprivation on mTOR kinase, but many protein kinases are involved in the regulation of cell proliferation. One of the ways to simultaneously screening the activity of several protein kinases, is to perform a protein kinase antibody array, which consist of a nitrocellulose membrane on which several protein kinase antibodies (against the protein and its phosphorylated forms) are spotted. This will also elucidate how the downstream mTOR signaling targets are regulated.
RESPONSE 6: Our work is based on Rubin’s theory (ref 10 and 42) postulating that among the kinases involved in proliferative pathways, mTOR has a very high Km for MgATP and could be very “sensitive” to Mg deficiency. In this paper we limited our interest to mTOR, but this a good suggestion. Actually, we have begun experiments in order to explore the effect of Mg deprivation on the signaling cascade(s) involving mTOR, but this requires a large body of work. However, this point is important, and a sentence has been added in the discussion (lines 463-465)

Reviewer 2 Report
Manuscript ID: nutrients-1163422
Title: Magnesium and osteosarcoma cell proliferation.
In this study, authors demonstrate that Magnesium deficiency inhibits the proliferation of SaOS-2 osteosarcoma cells ant this response may be associated cell cycle arrest and mTOR hypophosphorylation at serine 2448. This study shows interesting findings using a variety of good experimental techniques. However, there are many points to need to be improved and clarified by authors to increase the potential of suitability for publication.
- Title of this manuscript is too vague. The current title may be understood in the context of the manuscript, but it is necessary to be written more specifically.
- In the study, authors used only 1 mM as the concentration of magnesium (MgCl2) and showed only 24 h treatment results or only 24 h and 48 h treatment results in each test. Haven’t the authors tested those at other time points and concentrations? Was toxicity observed at other time points and at other concentrations? It is necessary to logically rationalize the reasons for determining the test conditions (time, concentration etc.) that authors have chosen. Such rationalization can support a clear interpretation and understanding of the findings of the reviewer and future readers. Therefore, their rationalization should be described by authors.
- In the same context, in the Results section (L206-210) authors described that SaOS-2 osteosarcoma cells were grown for 24 h and 48 h in a medium containing 5% dFBS in presence or absence of 1 mM MgCl2. In the Fig. 1, authors presented the intracellular content of total magnesium in SaOS-2 cells grown for only 24 h. What was the result for SaOS-2 cells grown for 48 h?
- In Fig. 2B, cell sizes in the deprivation of 1 mM MgCl2 show to be larger compared to those in the presence of 1 mM MgCl2. As mentioned by authors, magnesium is an essential nutrient for normal physiological functions of cells. No unusual changes in the cells, including morphological changes, were found due to the deprivation or addition of MgCl2? For a clear interpretation of the results, authors need to provide the results of morphological observations of cells under test conditions as well as normal condition.
- In the Results section, authors described that the deprivation of MgCl2 did not induce any significant increase of LDH release at 24 h and 48 h. However, at 48 h, LDH release seems to be quite increased by the deprivation of MgCl2. The reviewer would like to know what the p-value was.
- In Fig. 5A (right graph), the graph is too confusing for interpretation of the results. In its legend, authors describe that “the levels of mTOR and phosphorylated mTOR in the presence of Magnesium are arbitrarily taken as 1”. However, there is a discrepancy between the graph and its legend. This issue should be resolved clearly.
- Fig. 5B: The expression levels of LC3-II and LC3-I and the conversion ratio of endogenous LC3-II to LC3-I are widely known to be important markers of autophagy. In the Results section, authors described that magnesium deficiency did not induce LC3 cleavage and, consequently, did not increase the level of LC3-II protein, the cleaved and lipidated form of LC3 protein. Based on the results, in the Discussion section, authors suggested that in SaOS-2 cells, magnesium deficiency leads to reduced proliferation without affecting autophagy. In upper panels, magnesium deficiency seems to change LC3-I and beta-actin expression. Given the changes in LC3-1 and beta-actin expression, it is not sufficient to determine the effect of magnesium on autophagy based on the present results. Therefore, to clarify autophagy events, it is necessary to conduct additional autophagy test or to analyze the results of the conversion ratio of endogenous LC3-II to LC3-I after normalization to beta-actin. On the basis of this results, authors should present the result as a statistical graph. In addition, in lower panel the authors should provide bands of loading control protein.
- The present manuscript needs more careful and organized description of data analysis, authors’ opinion, and figure legends.
- Fig. 3A, legend: The ** mark is not shown in the graph.
- Fig. 3B: The unit mUE/ml in the title of the x-axis in Figure 3B should be checked if it’s correct.
- Since there are many "slips" in the manuscript, careful language checking is needed (L82, L101, L250, L252, L389 etc.).
- In this manuscript, decimal points should be changed to ‘.’.
- Abbreviations should be used after providing their full name (spelling).
- There are units that can be confusing. It is necessary to use unit in a certain form (i.e., min, minutes, h, hour(s), /ml, /mL, mL-1 etc.). In addition, the unit display of centrifugation force needs to be corrected.
- In the Materials and Methods section, authors need to provide information about the source of the human osteosarcoma cell line SaOS-2. The source of antibodies used for western blotting analysis and their dilution rate should be presented.
- What does ‘0,45 μ filter’ mean (in L101)?
- What does ‘240 g x 10 min’ mean (in L160)?
Author Response
Response to Reviewer 2 Comments
Major
POINT 1: Title of this manuscript is too vague. The current title may be understood in the context of the manuscript, but it is necessary to be written more specifically.
RESPONSE 1 :We changed the title and now it is more specific.
POINT 2: In the study, authors used only 1 mM as the concentration of magnesium (MgCl2) and showed only 24 h treatment results or only 24 h and 48 h treatment results in each test. Haven’t the authors tested those at other time points and concentrations? Was toxicity observed at other time points and at other concentrations? It is necessary to logically rationalize the reasons for determining the test conditions (time, concentration etc.) that authors have chosen. Such rationalization can support a clear interpretation and understanding of the findings of the reviewer and future readers. Therefore, their rationalization should be described by authors.
RESPONSE 2: Magnesium concentration of 1 mM is the standard amount present in all commercial media, in agreement with the literature reporting the value of extracellular Magnesium concentration.
The Rubin’s hypothesis on cell proliferation locates the crucial role of magnesium in early events when growth factors are typically activated. Therefore, we focused our attention at 24 and 48 hours. We modified the text to better clarify this rationale(305-308).
POINT 3: In the same context, in the Results section (L206-210) authors described that SaOS-2 osteosarcoma cells were grown for 24 h and 48 h in a medium containing 5% dFBS in presence or absence of 1 mM MgCl2. In the Fig. 1, authors presented the intracellular content of total magnesium in SaOS-2 cells grown for only 24 h. What was the result for SaOS-2 cells grown for 48 h?
RESPONSE 3: As specified above, in the previous point, we were primarily interested on early events of cell proliferation.
POINT 4: In Fig. 2B, cell sizes in the deprivation of 1 mM MgCl2 show to be larger compared to those in the presence of 1 mM MgCl2. As mentioned by authors, magnesium is an essential nutrient for normal physiological functions of cells. No unusual changes in the cells, including morphological changes, were found due to the deprivation or addition of MgCl2? For a clear interpretation of the results, authors need to provide the results of morphological observations of cells under test conditions as well as normal condition.
RESPONSE 4: Although cell size in the deprivation of 1 mM MgCl2 looks like larger compared to that in the presence of 1 mM MgCl2, the 2D cell shape cannot be taken as measure of size lacking in the third dimension (thickness). To overcome this issue, we measured the cell thickness by the Atomic Force Microscopy. Specifically, the volume of the two cells showed in Fig.2B was 110 µm3 for the SaOS-2 in the presence of magnesium and 127 µm3 for the SaOS-2 in absence of magnesium. Finally, we remark that the volume has been calculated in fixed cells avoiding any other consideration about the morphology of the two populations of cells.
POINT 5: In the Results section, authors described that the deprivation of MgCl2 did not induce any significant increase of LDH release at 24 h and 48 h. However, at 48 h, LDH release seems to be quite increased by the deprivation of MgCl2. The reviewer would like to know what the p-value was.
RESPONSE 5: Thanks for the careful review. We fully agree: figure 3B is awkwardly wrong. The standard deviation bars were incorrectly reported. We corrected the graph and reported the statistical analysis showing that differences were not significant.
POINT 6: In Fig. 5A (right graph), the graph is too confusing for interpretation of the results. In its legend, authors describe that “the levels of mTOR and phosphorylated mTOR in the presence of Magnesium are arbitrarily taken as 1”. However, there is a discrepancy between the graph and its legend. This issue should be resolved clearly.
RESPONSE 6: Thanks for the observation. We corrected the caption of figure 5A accordingly.
POINT 7: Fig. 5B: The expression levels of LC3-II and LC3-I and the conversion ratio of endogenous LC3-II to LC3-I are widely known to be important markers of autophagy. In the Results section, authors described that magnesium deficiency did not induce LC3 cleavage and, consequently, did not increase the level of LC3-II protein, the cleaved and lipidated form of LC3 protein. Based on the results, in the Discussion section, authors suggested that in SaOS-2 cells, magnesium deficiency leads to reduced proliferation without affecting autophagy.
In upper panels, magnesium deficiency seems to change LC3-I and beta-actin expression. Given the changes in LC3-1 and beta-actin expression, it is not sufficient to determine the effect of magnesium on autophagy based on the present results. Therefore, to clarify autophagy events, it is necessary to conduct additional autophagy test or to analyze the results of the conversion ratio of endogenous LC3-II to LC3-I after normalization to beta-actin. On the basis of this results, authors should present the result as a statistical graph. In addition, in lower panel the authors should provide bands of loading control protein.
RESPONSE 7: We agree that LC3 expression alone is not sufficient to detect autophagy. As required by the reviewer, figure 5 has been revised and now it depicts the results of western blotting as graphs reporting the densitometric analysis after normalization to beta-actin. The figure caption has been updated and a sentence about normalization to beta-actin has been added in the material section (line 167). Furthermore, the main text has been slightly changed in the results section (lines 381-383) and in the discussion (lines 451-452).
POINT 8: The present manuscript needs more careful and organized description of data analysis, authors’ opinion, and figure legends.
RESPONSE 8: Thanks for the suggestions: we have carried out a careful review
POINT 9: Fig. 3A, legend: The ** mark is not shown in the graph.
RESPONSE 9:We corrected the figure.
POINT 10: Fig. 3B: The unit mUE/ml in the title of the x-axis in Figure 3B should be checked if it’s correct.
RESPONSE 10: We corrected the figure.
POINT 11: Since there are many "slips" in the manuscript, careful language checking is needed (L82, L101, L250, L252, L389 etc.).
RESPONSE 11: Done.
POINT 12: In this manuscript, decimal points should be changed to ‘.’.
RESPONSE 12 Done.
POINT 13: Abbreviations should be used after providing their full name (spelling).
RESPONSE 13: We carefully checked in the text and verified that abbreviations are used after that their full name is provided.
POINT 14: There are units that can be confusing. It is necessary to use unit in a certain form (i.e., min, minutes, h, hour(s), /ml, /mL, mL-1 etc.). In addition, the unit display of centrifugation force needs to be corrected.
RESPONSE 14: Done.
POINT 15: In the Materials and Methods section, authors need to provide information about the source of the human osteosarcoma cell line SaOS-2. The source of antibodies used for western blotting analysis and their dilution rate should be presented.
RESPONSE 15: Thank you for suggestion. We specified the source of the human osteosarcoma cell line SaOS-2(lines 107-108). Moreover, we indicated the source of antibodies used for western blotting analysis and their dilution rate (lines 164-166)
POINT 16: What does ‘0,45 μ filter’ mean (in L101)?
RESPONSE 16: We clarified that it means 0.45 μm pore size membrane filter (line 104).
POINT 17: What does ‘240 g x 10 min’ mean (in L160)?
RESPONSE 17: We rewrote it in the correct form.

Reviewer 3 Report
In this manuscript, using SaOS2 osteosarcoma cells as a model, the authors measured intracellular magnesium concentration and distribution, and studied cell response to magnesium starvation at cellular as well as molecular levels. The authors found that the magnesium was mainly localized near the plasma membrane in magnesium starved cell. Moreover, the authors found magnesium depletion induces cell cycle arrest at G0/G1 phase through upregulation of p27 in protein level and downregulation of mTOR pathway activity. The methodologies are adequate and the manuscript is well presented in general. The reviewer has a few concerns about the manuscript.
- RNAseq analysis will provide an overall picture of altered pathways at gene expression level. Is that possible to provide RNAseq analysis for magnesium starvation in SaOS2 cells?
- Figure 1, intracellular Magnesium levels are presented as “nmol/106”. For the convenience of comparison, can the authors measure the volume of SaOS2 cells and then calculate the concentration of magnesium using “mM”.
- Figure 2B, do the numbers indicate the highest or average magnesium concentrations? Please double check the if the scales are correct and clear (especially the powers of 10power).
- Figure 4D, please label the percentage of p27 positivity cells for each panel.
- Lines 400-401, please double check the grammar.
- Can the authors change the title to a more specific one reflecting the main conclusion of the manuscript?
Author Response
Response to Reviewer 3 Comments
POINT 1: RNAseq analysis will provide an overall picture of altered pathways at gene expression level. Is that possible to provide RNAseq analysis for magnesium starvation in SaOS2 cells?
RESPONSE 1 : We agree that RNAseq analysis would provide an overall picture of altered pathways at gene expression level. However, this would require a specific study with a large body of work, which is outside the scope of this paper.
POINT 2: Figure 1, intracellular Magnesium levels are presented as “nmol/106”. For the convenience of comparison, can the authors measure the volume of SaOS2 cells and then calculate the concentration of magnesium using “mM”.
RESPONSE 2: We added in the main text the mM concentrations of magnesium (lines 229-234) using the volume of same cells reported in our previous work (ref. 37).
POINT 3: Figure 2B, do the numbers indicate the highest or average magnesium concentrations? Please double check the if the scales are correct and clear (especially the powers of 10power).
RESPONSE 3 The values in fig 2B indicate the real intracellular Mg concentration of the two whole cells. We checked and clarified the 10power in the scale bar.
POINT 4: Figure 4D, please label the percentage of p27 positivity cells for each panel.
RESPONSE 4: We added the percentage of p27 positive cells in each panel and in the main text (lines 350-351).
POINT 5: Lines 400-401, please double check the grammar.
RESPONSE 5: Done.
POINT 6: Can the authors change the title to a more specific one reflecting the main conclusion of the manuscript?
RESPONSE 6: Thank you for your suggestion: we changed the title to a more specific one, reflecting the conclusions.

Round 2
Reviewer 1 Report
The paper is acceptable in the present form, with some minor text editing.
Author Response
Thank you so much for your review and for your valuable suggestions that
have helped us improve our manuscript.
We have made some small changes to the text to further clarify the choice
of experimental times and improve the manuscript.
Reviewer 2 Report
Manuscript ID: nutrients-1163422
Title: Magnesium and osteosarcoma cell proliferation
Authors: Concettina Cappadone1 et al.
The revised manuscript has been improved a lot. The authors responded almost adequately to the comments or suggestions of the reviewer and modified the manuscript. The current version of the manuscript may be almost sufficient for publication. However, there are still a few points that need to be clarified and improved.
- In the Result section, authors described that the medium was replaced with a medium containing 5% dFBS in presence or absence of 1 mM MgCl2 and grown for 24 and 48 h in order to study the effects of magnesium deficiency in human osteosarcoma SaOS-2 cells. In Figure 1A (entitled cell viability), authors observed that the effects of magnesium deficiency on the intracellular total magnesium contents in human SaOS-2 osteosarcoma cells at only 24 h incubation. It needs to describe the rationalized reason for determining the test time (24 h) to assess the total intracellular magnesium content, as done in proliferation test results section. Otherwise, authors may also consider putting the total magnesium content and imaging results section as the next section of the proliferation results.
- In the Result section, authors described that considering the SaOS-2 volume, the intracellular magnesium content was 13.7 mM in cells grown with magnesium and 8.4 mM in cells grown without magnesium. For Fig. 2B, authors described that magnesium concentration was estimated to be 87 mM in cells grown in the presence of MgCl2 and decreased to 26 mM in magnesium-deprived cells. How did the magnesium content be estimated, taking into account the volume of SaOS-2 cells? Is it the average of all cells or that of a single cell? It needs to be clearly described. In addition, a method of calculating the magnesium content by considering the volume of cells needs to be described in the method section.
- Authors considered cell volume alternation in intracellular magnesium content analysis. If so, cells grown without magnesium seems to be under abnormal physiological condition. Did authors confirm that increased volumes in cells grown without magnesium were recovered by incubation in normal medium? This issue is very important. Authors should be discussed the issue in the Discussion section.
- Fig. 5C shows that LC3-II expression is increased in cells incubated with 10 mM chloroquine (Clq) for 24 h. Was the increased level not significant compared to the control group? In addition, it needs to describe for Fig. 5C in the Results section.
- In this manuscript including figure legends, decimal points should be changed to ‘.’.
- Page 7 of 15, section 3.2: In 2nd paragraph, it need to provide a literature citing the Rubin’s hypothesis.
